# Assessing Physiological and Behavioral Stress Parameters in Trained Goats During Repeated Blood Sampling

**DOI:** 10.3390/ani16010105

**Published:** 2025-12-30

**Authors:** Jennifer Meier, Hildegard Just, Matthias Steinfath, Carola Fischer-Tenhagen

**Affiliations:** 1Department Experimental Toxicology and ZEBET, German Federal Institute for Risk Assessment (BfR), 10589 Berlin, Germany; jennifer.meier@posteo.de (J.M.); matthias.steinfath@bfr.bund.de (M.S.); 2Department Safety in the Food Chain, German Federal Institute for Risk Assessment (BfR), 10589 Berlin, Germany; hildegard.just@bfr.bund.de

**Keywords:** eye temperature, refinement, ruminants, severity assessment, positive reinforcement training, venipuncture

## Abstract

Blood sampling is a common procedure in veterinary care and animal research, but it can cause stress due to accompanying handling and pain. When animals repeatedly experience this procedure, stress may increase if they anticipate discomfort or decrease if they become accustomed to the routine or receive positive training. This study describes how repeated blood sampling affects stress parameters in goats that were trained to cooperate and handled gently. Eight dairy goats had blood drawn once a week for over four months. The stress parameters were assessed by measuring changes in serum cortisol concentrations, eye temperature, and behavior. The results showed that serum cortisol levels and eye temperature did not increase over time. However, the goats approached the blood sampling area faster and more voluntarily as the study progressed. In this study, repeated blood sampling did not affect stress response in goats. This information is valuable for improving animal welfare in research and veterinary care, showing that careful handling and positive training can help reduce the negative impact of routine procedures on animals’ well-being.

## 1. Introduction

Stress in animals is a complex physiological reaction with far-reaching implications for animal welfare, scientific research validity [1], and ethical considerations [2]. Stress refers to the physiological and psychological responses to environmental challenges. Stress is differentiated as eustress and distress [3]. Eustress describes a positive emotional state associated with activation and engagement. Distress is a destructive type of stress and reflects a state of negative emotional arousal associated with dissatisfaction and disengagement [4]. In contrast to eustress associated with successful coping strategies, distress occurs when animals face a stressor exceeding their coping resources. Affective states in animals can be further described by a dimensional system of valence (positive or negative) and arousal (high or low) [5].

In animal experiments under the framework outlined in Annex VIII of the EU 2010/63 directive, distress is relevant for classification of the experiment in the four categories of severity, i.e., “non-recovery”, “mild”, “moderate”, and “severe”. These categories help researchers to assess the severity of their experiments and to address the impact on the welfare of laboratory animals. In the EU in 2022, almost half (3,921,024) of all animals used in experiments experienced stress of mild severity [6]. Examples of procedures that are assessed with mild severity include administration of anesthesia, non-invasive imaging, or short-term deprivation of a social partner (Directive 2010/63/EU). Stress of mild severity can still result in short-term stress for an animal, impacting animal well-being. For example, even routine procedures such as blood sampling, as a part in many research protocols, may cause stress in animals [7].

Blood sampling, or venipuncture, is a standard procedure in animal experiments, e.g., for the detection of metabolic changes [8] or environmental influences on the animal’s homeostasis [9]. Apart from experimental studies, “practices [that are] not likely to cause pain, suffering, distress or lasting harm equivalent to, or higher than, that caused by the introduction of a needle in accordance with good veterinary practice” can serve as a threshold for determining whether a procedure falls under the scope of EU Directive 2010/63 and is therefore classified as an animal experiment requiring approval [10]. In farm animals such as goats, blood sampling is commonly accompanied by restraining of the animal. Therefore, not only the short/mild pain caused by punctuation of the skin but also restraining and short-term isolation may cause stress in the animal [11,12]. When this procedure is repeated several times during an ongoing experiment, negative experiences can aggravate the affective state of animals [13,14] or human patients [15] in this situation. In contrast, repetition of a stressful procedure, such as transportation, can reduce stress through habituation [16]. This dichotomy also occurs in the severity assessment framework set forth by the European Commission, as it mentions that “there is no direct link between frequency and severity. […] When interventions are repeated, there is the potential for acclimatization, which may reduce severity, […]. Conversely, repetition may increase severity, e.g., due to anticipation of a stressful procedure […]” [17]. In conclusion, it is likely that lower severity could occur when the acclimatization phase is optimized while the sensation of a procedure is minimized.

Assessment of the stress accompanying certain procedures involved in an experiment helps to categorize the severity of the experiment. Stress can be assessed using physiological and ethological parameters. Hydbring-Sandberg et al. [18] highlighted that stress measurement must be multifaceted to allow a reliable prediction. They measured and compared the stress reaction in goats in different handling situations (loose holding vs. tethering). The serum cortisol concentration did not differ between groups, but other physiological parameters, such as blood pressure or heart rate, were higher in the tethered group. However, the measurement of glucocorticoids as a crucial part of the hypothalamus–pituitary–adrenal (HPA) axis is frequently used for stress evaluation [11,19]. Cortisol, a glucocorticoid, serves as a reliable biomarker of stress reactions, yet its measurement may present challenges in interpretation and standardization, especially when it comes to the assessment of animal welfare [20]. Infrared thermography is promoted as a non-invasive method to evaluate stress [21,22,23] or as an indicator of the emotional state of farm animals, as described in sheep [24] or cows [25]. Through activation of the HPA axis and the sympathetic nervous system, peripheral vessels are constricted. This leads to a decrease in peripheral skin temperature, e.g., on the ears of rabbits [26], as well as an increase in eye region surface temperature [27]. Accordingly, Ref. [22] found eye infrared thermography to be a useful immediate and non-invasive physiological tool to assess stress response in sheep. This is in accordance with the study of Bartolomé, Azcona, Cañete-Aranda, Perdomo-González, Ribes-Pons, and Terán [23]. They described eye thermography as an appropriate and non-invasive tool to explore stress levels in goats during routine management procedures.

Ethograms standardize the recording of behavior, allowing results to be compared across observers, periods, and studies [28]. In addition to physiological assessment, observing stress-related behaviors provides valuable insights into animal welfare and emotions when not only the level of arousal but also the valence of such emotions is investigated [5]. This is particularly relevant, as it is difficult to discriminate the valence of arousing situations such as an aversive stimulus (e.g., isolation) and a pleasant stimulus (e.g., food reward). Therefore, behavior responses, such as latency in locomotion [29] vocalization [30], or escape attempts [31,32], can help to assess the emotional state of the animal. Siebert et al. [31] observed that completely isolated goats exhibited lower activity levels, reared less frequently, and emitted fewer high bleats compared to partially isolated animals. In contrast, the latter showed more active behaviors such as increased locomotion, rearing, jumping, and high bleats, as well as motivated attempts to re-establish social contact

To our knowledge, there is a lack of information regarding the affective state of goats undergoing repeated blood sampling over a longer period. Filling that gap could help to refine severity assessment for animals in experimental settings. Therefore, this study observes the influence of repeated blood sampling on the affective state of the animals. We wanted to observe how repeated blood sampling would affect stress parameters in goats that underwent an intensive cooperative vet care training protocol and were handled with a focus on gentle handling

## 2. Materials and Methods

The data were collected as part of a study on the transfer of per- and polyfluoroalkyl substances (PFAS) in hay to milk from dairy goats (transfer study; [33]). This study was approved by the Competent Authority (Landesamt für Gesundheit und Soziales Berlin—LAGeSo) under the reference number G0050/22 (date of approval: 12 September 2022).

### 2.1. Animals and Housing

Ten lactating German Improved White female goats (*Capra aegagrus hircus*; first to fourth lactation, aged 1–4 years) were purchased from a commercial dairy goat farm and were included in the transfer experiment. They were housed at the German Federal Institute for Risk Assessment (BfR) for a period of 24 weeks (24 April 2023 to 9 October 2023). The goats were divided into two groups of five and housed in two separate pens under matched conditions. Each pen contained the housing pen, a training area, a hay storage area, and a treatment area (Figure 1). The housing pen consisted of a floor area of 25 m^2^ and was enriched with deep straw bedding and four to five platforms for the goats to jump on or seek shelter under. On Weeks 5, 10, and 13, the housing pen was subdivided with mobile fencing to house the goats individually and to collect goat-specific feces and urine samples for the PFAS transfer study. Data of four goats from each group (*n* = 8) were used in this study, excluding the two surrogate animals, which were used as backup. They were not used for blood sampling and data collection but remained in the group during the entire experimental period. No goat had to be excluded during this study.

Goats were given ad libitum access to hay in slow feeders (HeuToy, Udo Röck GmbH, Bad Saulgau, Germany), water, and mineral stones. Each goat received milk performance-oriented concentrate feed (M 18-4 Green pelleted, Agravis Ost GmbH & CO.KG, Fürstenwalde, Germany) tailored to its milk yield and was offered additional feed during training or handling procedures as a reward. During the transfer study, five goats were consistently fed PFAS-free hay throughout the study (Group C), while the remaining five goats were exposed to PFAS-contaminated hay for a period of time (Week 5–12; Group E). All goats were checked daily for any signs of decline in general condition. No discernible adverse effects on the general health status of the goats were observed during the study. However, Group E exhibited, on average, a significantly lower body weight (during the complete period) and lower milk yield (during PFAS contaminated hay exposure period) compared to those of Group C.

The treatment pens (one in each barn, see Figure 1) were located opposite the housing pens and measured approximately 12 m^2^ in size. This area contained a milking parlor with a portable milking machine (milk trolley for goats, Milkline, Italy), a table, and on blood-sampling days, a black rubber mat measuring 85 × 45 cm (floor target). Goats were individually let into the treatment pen for milking (1× daily at 1 p.m.), blood sampling (1–3× weekly), and positive reinforcement training (PRT). During milking, each goat received concentrated feed adjusted to milk yield.

### 2.2. Training

The ethical approval according to 2010/63/EU included a training protocol to prepare all goats for the blood sampling as a refinement measure, as part of the 3R concept to reduce stress during an experiment [34]. Therefore, after two weeks of habituation, all goats underwent positive reinforcement training to enhance voluntary behavior. In this study, PRT serves as an umbrella term for positive reinforcement training methods such as cooperative vet care, husbandry training, or clicker training and shaping, when used to refine experimental procedures. Before arrival at the BfR, the goats had not been trained. The training started in Week 3 and continued until Week 17, conducted by J.M. J.M. had 2 two years of specific experience in training goats using PRT in an experimental context. On average, each goat underwent eight training sessions of 13 min each. Differences in the number and duration of sessions between the two groups resulted from differences in learning pace within the goats. The training followed the principles of PRT, which means that animals receive a reward (food) for a desired behavior to enhance the probability that the rewarded behavior is repeated [35]. The initial phase of training was used to habituate both groups in the housing pen to accept food from the hand. This was achieved after 120 min in Group C (in four sessions) and 105 min (in three sessions) in Group E. The next goal was to train each goat to follow a hand as a leading target (hand target) in the training area. For this, the goat had to touch the trainer’s presented hand with its nose. The distance to the trainer’s hand was increased during training as a low-stress method of guiding the animal to, for example, the training area. All goats learned to follow the hand target for at least 5 m with minimal distraction. The floor target, represented by a black rubber mat (85 × 45 cm), was then introduced. Training with the floor target was conducted in both the training and the treatment areas. The goats were trained to stand with all four hooves on the target and maintain this position with minimal distraction. This helps to place the goats in a good position for handling with minimal physical restraint. The goats were then trained to tolerate a hand under the chin, a crucial position for blood sampling. The number of training sessions and the performance of each goat are listed in Table 1. Achievements during the training sessions included following a hand target for at least 5 m (hand target), finding a position on floor target (with or without a supporting signal, such as a hand target), voluntarily placing the chin on the trainer’s hand, and tolerating touches.

### 2.3. Blood Sampling

In the context of the PFAS study, blood samples were collected on Mondays, Wednesdays, and Fridays during Weeks 5, 6, 10, 11, 13, and 14 and once a week on Wednesdays during all other weeks of the experiment, with a total of 33 blood samplings per goat over the entire duration of the experiment. They were collected from the jugular vein (alternate sides) in the treatment area (Figure 1) using a BD Vacutainer Safety-Lok cannula (21G, 0.8 × 19mm) and BD Vacutainers (Becton Dickinson AG, Allschwil, Switzerland). Blood sampling was performed as follows: the goat was led to the treatment area and onto the floor target using hand targeting/luring or, if necessary, using light force on a leash. While the goat was being rewarded in this position, a caretaker stepped to the left side of the goat and lightly restrained the goat with their leg next to the goat’s chest, with one hand under the chin and the other hand, if necessary, on the goat’s head. A second person knelt beside the goat’s left shoulder to compress the goat’s jugular vein by hand. A third person (H.J.) disinfected the fur and skin over the jugular vein with alcohol and punctured the vein with the cannula. The goat was rewarded the moment the cannula exited the vein. While the goat was receiving the reward, the caretaker released the chin restraint. The vein was then compressed with gauze at the site of the previous cannula insertion. If the vein did not bleed, the staff member removed the gauze and stepped back from the goat. After a final reward, the goat was returned to the pen.

To evaluate stress levels during blood sampling, the following parameters were assessed: (1) blood serum cortisol concentrations, (2) temperature change at the inner corner of the eye, and (3) behavior, observed using an ethogram. All parameters were recorded at nine designated time points throughout the study (*time points* 1–9 correspond to Wednesdays in study Weeks 6, 7, 10, 12, 13, 15, 17, 21, and 23).

#### 2.3.1. Blood Serum Cortisol

Blood serum samples were analyzed at the Federal Research Institute for Animal Health, Friedrich-Loeffler-Institut (FLI), Braunschweig, Germany, and serum cortisol concentrations were determined using a commercial ELISA kit (Enzo Life Sciences, Farmingdale, NY, USA). Three 10 mL BD Serum Vacutainers (Becton Dickinson GmbH, Heidelberg, Germany) were centrifuged at 2100 rpm for 15 s, and the serum was pipetted. Aliquots of about 2 mL were stored at −21 °C until analysis. In addition to the nine specified time points, the zero sample of the transfer study (Week 2) was included for analysis. When the zero sample was taken, training with the goats had not yet started.

#### 2.3.2. Eye Temperature

Eye temperature images were captured using an infrared thermography (IRT) camera (VarioCAM HD, InfraTec, Dresden, Germany) and analyzed using IRBIS3 plus thermography software (version 3.1, InfraTec, Dresden, Germany). According to Sutherland et al. [36], the images were captured from a distance of approximately 50 to 100 cm at an angle of approximately 90 degrees to the left eye of the goat. With 25 frames per second, five images were captured for each image set. The first image in focus, with the eye open, was used for measurement. The maximum temperature was measured in the area of the medial canthus (see Figure 2). Two image sets per animal per *time point* were captured: the first with the restrained goat approximately 3 s before the skin puncture (T1) and the second approximately 5 s after the puncture during blood sampling (T2). The difference T2–T1 was used for statistical analysis. Relative humidity (%) and ambient temperature (°C) were measured before the procedure.

#### 2.3.3. Behavior

The blood sampling procedure was continuously recorded per group using a GoPro 7 White device (GoPRo, San Mateo, CA, USA). Behavior was evaluated retrospectively from video material. The recording started with the first goat leaving the pen and ended when the last goat exited the treatment area. The videos were then evaluated by JM using an ethogram consisting of the four behavioral characteristics of *latency*, *compliance*, *escape behavior* and *defensive behavior*, defined and characterized in Table 2. These indicators were selected due to their clear observability in the video material and their relevance in the goat’s behavior repertoire observed by the experimenters in the past. Other potentially relevant behaviors, such as trembling, ear and tail position, or vocalization [32], were either too subtle, too dependent on audio quality, or potentially restricted during the blood sampling procedure to be measured with consistent reliability. The BORIS [37] program was used for the ethogram evaluation.

### 2.4. Statistical Analysis

Data analysis and visualization were performed using R 4.2.2 [38] and Excel [39]. The potential predictor variables and the outcome variables are listed in Table 2. Linear models and linear mixed models were calculated to assess the effect of *time point* on the variables *cortisol*, *thermo*, and *latency*. These models consist of *cortisol*, *thermo*, and *latency* as response or dependent variables, *time point* as an independent continuous variable or a fixed factor, *humidity* and *temperature* as possible additional fixed factors, and goat as a random factor; *pen*, *person 1*, and *person 2* were included as possible additional random factors (see Table 2). We compared the Akaike Information Criteria (AIC) across these different models for each of the three continuous dependent variables. The model with the lowest AIC value was selected as the most suitable and therefore, the optimal model (using the lme function from the nlme package (version 3.1-160) in R and the lm function from the R package stats 4.2.2).

For analysis of the binary outcomes *compliance* and *escape behavior*, a similar approach was used (see Table 2). However, instead of linear and linear mixed models, generalized linear models (glm) and generalized linear mixed models were applied (glmm). The R function glmmTMB from the package glmmTMB and glm from the package stats were used. For the ordinal variable *defensive behavior*, an ordered logistic regression with the R function polr from the MASS package was applied, with *time point* as independent variable (see Table 2).

## 3. Results

### 3.1. Blood Serum Cortisol

The cortisol concentration in the blood serum ranged between 1.02 and 75.49 ng/mL over all animals and time points (see Figure 3). The optimal model (*time point* and *temperature* as fixed effects and *goat* and *person* 2 as random factors) estimated a slope of m= −0.125 ng/mL/time point. The result at the end of the period in comparison to the cortisol level in the zero-sample estimated by the model is −0.003 ng/mL. There was no significant influence of the time *point* on *cortisol* (*F*(1, 30) = 1.01, *p* = 0.32, 95% confidence interval (CI) [–0.855, 0.604]).

### 3.2. Eye Temperature

The eye temperature differences between “before” (Z1) and “after” (Z2) the needle insertion (thermo) were between −0.7 and 2.3 °C. The average temperatures before and after insertion of the needle were 37.30 °C (before; SD= 0.497 °C) and 37.40 °C (after; SD= 0.544 °C). For the variable *thermo*, the optimal model consists of the *time point* as the only fixed effect and *Person 1* and *Person 2* as random variables (*F*(1, 2) = 93.34, *p* = 0.01, 95% CI [–0.075, –0.029] °C per day). The *p* value and CI for the difference in the change in eye temperature over time suggest that the difference decreases via habituation to the procedure. The temperature difference decreased between 0.32 °C and 0.6 °C over the measurement period. This means that the difference nearly disappeared at the end of the measurement period, indicating that there was no effect of the needle insertion on the eye temperature at the end of the measurement period (see Figure 4).

### 3.3. Behavior

The blood sampling sessions took 3 to 10 min from separation from the group to the end of blood sampling. A maximum of three goats per time point attempted to escape from the position. No goat exhibited this behavior more than three times during the experiment. Per time point, two to a maximum of five animals showed at least one of the defensive reactions. Goat 2 showed no defensive reactions at any time, whereas goat 8 showed defensive behavior at all times. For example, kicking behavior was observed 35 times, 25 of which were by goat 8. The number of goats voluntarily walking on the floor target varied from one (time point 1) to all (*n* = 8, time point 5). On average, they required 17.4 s (time point 6) to 51.3 s (time point 2) to enter the position (see Figure 5).

Within the optimal model for compliance (*time point*, temperature, and humidity as independent variables), there was a significant decrease in the number of goats that had to be forced to walk on the floor target (*z*(66) = −2.78, *p* = 0.005). For the variable *latency* (optimal model: linear model with *temperature* and time point as independent variables), a decrease of—3.41 s/time point was estimated. The decrease at the end of the period compared to the results for the zero sample is −31%. There was a significant decrease in time the goats required to step on the floor target (*F*(1, 64) = 26.03, *p* < 0.001, 95% CI [−4.92, −1.90] s per time point).

For the variable *escape behavior* (optimal model: generalized linear model with the time point as the independent variable), there is no clear indication of whether the value of the indicator is increasing or decreasing (*z*(71) = −1.17, *p* = 0.24). For the variable defensive behavior, no significant increase or decrease was found over the time points (*z*(69) = −0.89, *p* = 0.37). The Pearson correlation between cortisol and latency is low (*r* = −0.055; *t*(65) = −0.44, *p* = 0.66). Moreover, there is no statistically significant relationship between cortisol and compliance (*t*(65) = 0.98, *p* = 0.329) or cortisol and escape behavior (*t*(65) = −0.26, *p* = 0.795).

## 4. Discussion

The aim of the study was to observe stress parameters in goats subjected to repeated blood sampling. We did not see an increase over time in any stress parameter during the experiment. First, blood serum cortisol showed no significant change over time. A possible interpretation might be that repeated blood sampling did not lead to a higher stress level in terms of increased arousal. This and the model estimation of a slight decrease in cortisol concentration over time aligns with the findings of Andanson et al. [40]. In their study, the serum cortisol concentration of lambs (blood taken using catheters) were compared after saliva sampling, venipuncture, and no intervention. Handling was described as being performed “carefully”, and the animals were moved in a “gentle” manner for restraint for sample collection. On the first day, the group of lambs receiving venipuncture exhibited a significantly higher serum cortisol concentration compared to the results for saliva sampling or for the control group. After one week of blood sampling twice daily, no significant differences were found among all three groups. However, the serum cortisol concentrations in the venipuncture group were significantly lower compared to those on the first day of treatment, indicating that venipuncture induced the release of cortisol only at the initial exposure time and not after repeated sampling. Andanson et al. [40] attribute this to habituation. In our study, a plausible scenario could have been that habituation acted as a buffer to the stress, leading to no significant increase in serum cortisol concentrations. This is consistent with the EU guideline’s hypothesis that stress levels during a repeated intervention may decrease due to habituation and desensitization [17]. In our study, the procedure of blood sampling was quick, with 3 to 10 min from separation from the group to the end of blood sampling. Kannan et al. [41] found that when goats were isolated from the group, cortisol concentrations peaked immediately. However, Kruger et al. [11] found a peak in cortisol concentrations 10 min after handling stress. We may have observed a change in blood serum cortisol if an additional measurement had been taken 20–30 min after the first intervention. Due to the design of the underlying experiment, this was not feasible, and therefore, the results of the cortisol measurement must be regarded as preliminary and interpreted with caution. However, in this study, PRT and gentle handling may have contributed (while not significantly) to the tendency for decreasing cortisol concentrations. This would be consistent with the findings of other studies in which, for example, PRT with grizzly bears [42] or minipigs [43], as well as habituation in mice [44], led to a decrease in cortisol concentration during blood sampling. The positive effects of habituation and PRT may be difficult to disentangle, as both can occur at the same time.

Eye temperature showed no significant changes over time. This suggests that the act of puncturing the skin did not have an immediate, consistent effect on the peripheral vasculature. As shown by several studies, eye temperature increased after a stressor such as veterinary treatments [23], transportation [45], or shearing [22]. In contrast to the study by Arfuso et al. [22], the stress stimuli in our study may have been too mild, and the intervals between measurements too short, to induce a significant and consistent change in eye temperature. Therefore, thermographic measurements seem not to be suitable for assessing stress during short-term blood sampling procedures and could not address the research question of this study. In future studies on this topic, the baseline values of the goats before isolation from the group should be measured to assess the effects of handling and blood sampling compared to the results for neutral or relaxed states, without interventions. Here, baseline values were not considered necessary to assess the effects of the repetitive nature of these procedures and secondly, it was not feasible to measure these with a consistent angle and distance.

The ethological analysis of behavior during blood sampling did not reveal any relevant changes in regards to escape or defensive behavior. However, there were notable differences between individual animals. The defensive behavior “kicking” was primarily exhibited by one goat (no. 8) throughout all sessions. Accordingly, this goat showed the longest latency to arrive on the mat compared to that of the other goats at four different time points. The kicking behavior may have been unintentionally reinforced during the procedure, potentially contributing to its consistency. Despite this, the goat appeared attentive and engaged during training sessions and reached the mat voluntarily at the majority of time points. Beyond these individual, animal-specific effects, the force and technique of restraining influence the occurrence of this behavior as well. Importantly, the absence of defensive behaviors should not automatically be interpreted as indicating lower stress levels, as not showing such reactions may represent a freeze response within the four stress response patterns (fight, flight, freeze, or fawn). Therefore, no defensive behavior does not necessarily reflect low arousal or voluntariness, particularly under forceful or tight restraint. It remains unclear whether this is due to cooperation and voluntariness or to a passive stress response such as freezing [46]. However, a significant reduction in escape or defensive behavior with less force during restraining would have been expected with intensified training, since there are numerous examples in animal training, particularly in zoo animals, where voluntary participation in blood sampling is achieved, e.g., in macaques and chimpanzees [47] or lions [48]. Specifically, training to maintain the position for longer and under more distractions would have been required. Stimuli such as the touch of a second person, the presence of additional staff in the proximity of the animal, or the application of disinfectants at the injection site would have had to be incorporated into the training to facilitate the generalization of the learned behavior.

The significant decrease in the latency to reach the floor target argues against a possible anticipation of a negative stressor, as delayed locomotion is associated with fear in common behavioral tests (open-field test, novel-object test) [29]. In our study, the goats were specifically trained for this behavior. This is in line with the findings of Hutson [49], revealing that trained sheep entered a sheep-handling machine more quickly than untrained sheep. Furthermore, the goats more often stepped voluntarily onto the floor target and therefore, showed more compliance. Again, we presume that this is a training effect, as they were trained to step onto the floor target, with or without additional cues (such as the leading hand target). To determine the exact influence of PRT (and to differentiate from possible habituation effects) on the decreasing latency and increasing voluntary behavior, an untrained control group would have been required. The use of the floor target reduced the need for coercion in positioning. The decrease in time is in line with the findings of Sankey et al. [50], in which ponies trained with positive reinforcement showed increased interest in humans and approached them with a shorter latency than ponies trained with negative reinforcement. This highlights that PRT can improve the human–animal relationship. Furthermore, contact with calmer goats was perceived as pleasant by the staff and may therefore also contribute to a reduction of human stress, potentially improving safety in the workplace. Voluntary behavior in the blood sampling process may have resulted in a lower overall stress level for the goats. However, other parameters did not confirm this. For example, there was no significant correlation between cortisol concentrations and behavioral parameters, as well as no significant change in escape or aversive behavior. In the dimensional framework of animal emotions (valence and arousal) [5], behavioral measures such as latency to approach and voluntary behavior might be interpreted as potentially more positive—or at least less negative—valence [14]. However, because the animals in our study were explicitly trained and the physiological parameters did not suggest a significant change in arousal, this interpretation must remain cautious. Therefore, this study cannot address the emotional state of the goats during the procedures. Further behavioural parameters, such as vocalisation or positioning of ears or tail, as shown in the study by Briefer et al. [32], would have been necessary to do so.

We assume that three main factors may explain the stable stress parameters during repeated blood sampling in this study, the first being habituation and PRT. This is in line with the EU recommendation, which identifies habituation as a mitigating factor in repeated procedures [17]. Second is the optimized work procedures, such as a well-coordinated team, thorough preparation, and importantly, a calm approach to handling the animals. In addition to gentle handling, it is likely that the animals developed a positive expectation towards the trainer, who was present at all time points. In this study, goats were rewarded with food; therefore, classical conditioning took place via a positive association with the trainer, testing location, and procedures. A third reason may be that the stressor of blood sampling itself was reduced by choosing the smallest possible needle size to minimize the pain associated with the injection [51]. In the context of animal experimentation, the introduction of a needle is the threshold for a regulated animal experiment (Directive 2010/63/EU). However, there is evidence that even routine handling procedures, including subcutaneous vaccination, causes stress in goats [11]. As repeated blood sampling can be considered a relatively low-stress procedure for goats, the influence of repetition on other species or of more stressful procedures remains to be investigated. However, it is likely that with adequate PRT, habituation, and optimized protocols, repetition does not necessarily lead to increased stress levels.

A major limitation of this study is a missing control group. As this was a strictly descriptive study performed as a satellite in a PFAS transfer study, the ethical approval allowed only sufficient animals for this experiment but not extra animals for a control group. It was therefore not possible to include an untrained group of goats or a group that was included in only one or two time points of blood sampling. For the same reason, we were unable to increase the relatively small sample size. Furthermore, the measurement of further stress-relevant parameters like heart-rate-variability or blood pressure [18] was not possible, since the animals could not receive any other treatment than that approved for the transfer study. Thus, as our observations were made during a specific stressor (blood sampling), we cannot determine why the stress parameters showed no significant change over time; nevertheless, the measurements provide valuable information about the animals’ responses under these conditions.

Furthermore, ethical approval included all animals to be trained. Therefore, we could not evaluate the effect of training with a control group. Furthermore, the positive influence of PRT might have been more pronounced with a higher number of training sessions and an even more experienced trainer. However, even without full additional training and desensitization, many animals showed clear benefits, as they were noticeably calmer and more affiliative in their interactions with humans compared to the behaviors at the beginning of the study.

## 5. Conclusions

In summary, under well-controlled and positively reinforced handling conditions, repeated exposure to blood sampling as a potential stressful procedure does not necessarily influence stress parameters but appears to have a positive effect on the behavioral stress parameters assessed in this study. This observation can be attributed to three key factors: first, habituation and PRT for cooperative behavior and a positive human–animal relationship; second, optimal organizational practices, such as gentle handling, the use of minimally invasive needles, and a well-coordinated team; and third, the insertion of the needle for blood sampling as a relative mild aversive stimulus. If the first two factors are effectively implemented, it is possible that even more painful stimuli may not necessarily lead to an increase in stress over time. These observations highlight the potential benefits of habituation and PRT as refinement strategies but confirm that definitive conclusions require studies designed specifically to address this question, including control groups and repeated sampling within shorter intervals, to confirm whether habituation or other factors underlie the stable cortisol levels observed here.

## Figures and Tables

**Figure 1 animals-16-00105-f001:**
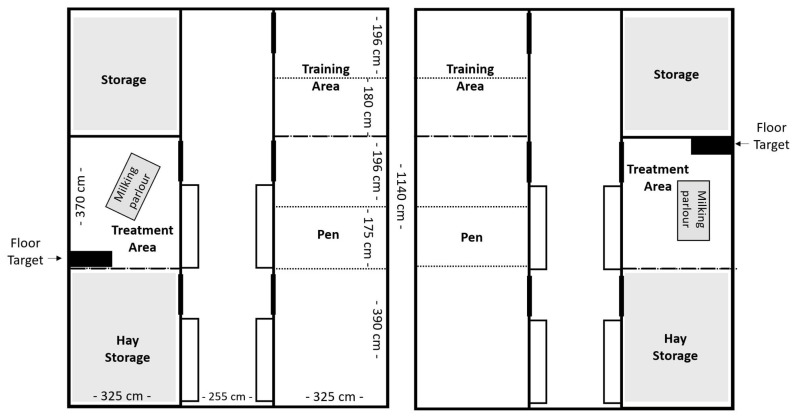
Pen design for Group E (**left**) and Group C (**right**). Dotted lines indicate fences, used only during weeks of individual housing.

**Figure 2 animals-16-00105-f002:**
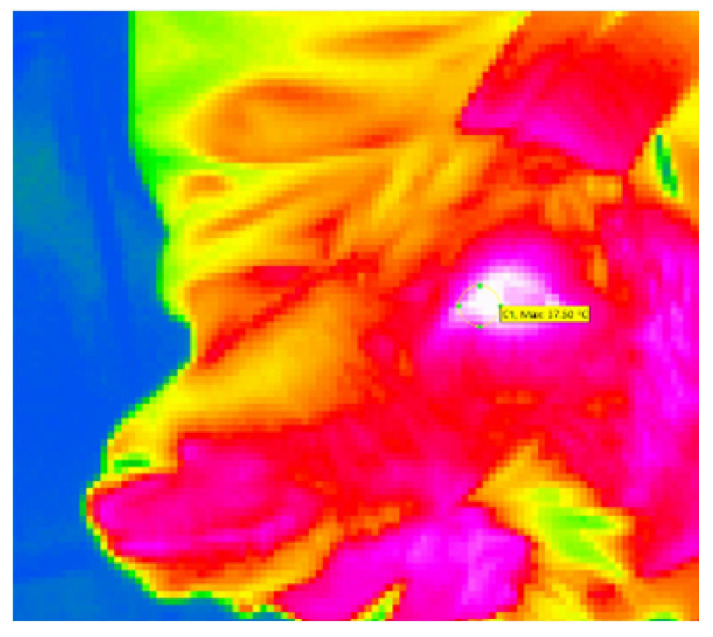
Thermographic image of a goat’s head, shot in lateral view during blood sampling. The circled area is the medical canthus of the animal’s left eye.

**Figure 3 animals-16-00105-f003:**
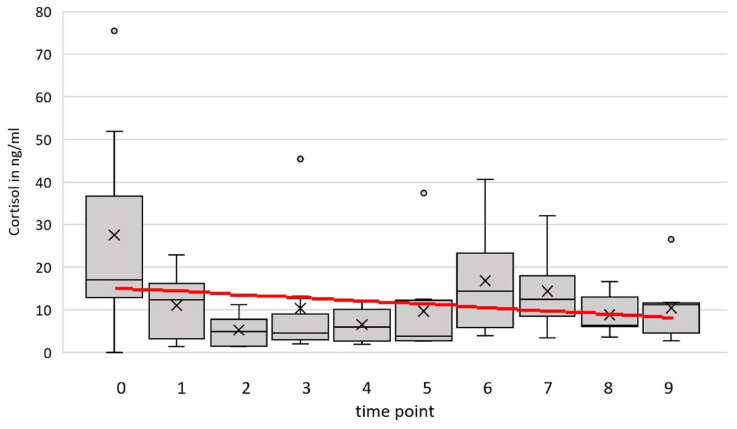
Cortisol level (ng/mL) of eight goats at nine time points and the additional zero sample (0). The red line represents the prediction by the simple linear model, with *time point* as the only independent variable.

**Figure 4 animals-16-00105-f004:**
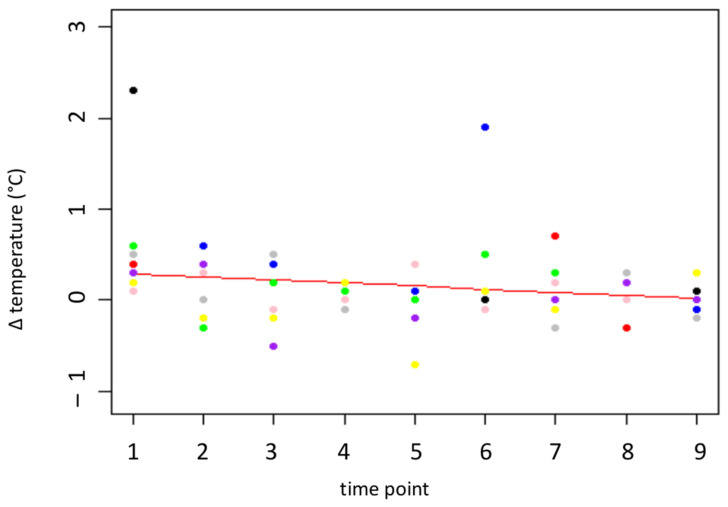
Temperature difference (°C) at the medial canthus before and after the insertion of the needle. The red line represents the prediction by the model, the colored dots represent the individual goats.

**Figure 5 animals-16-00105-f005:**
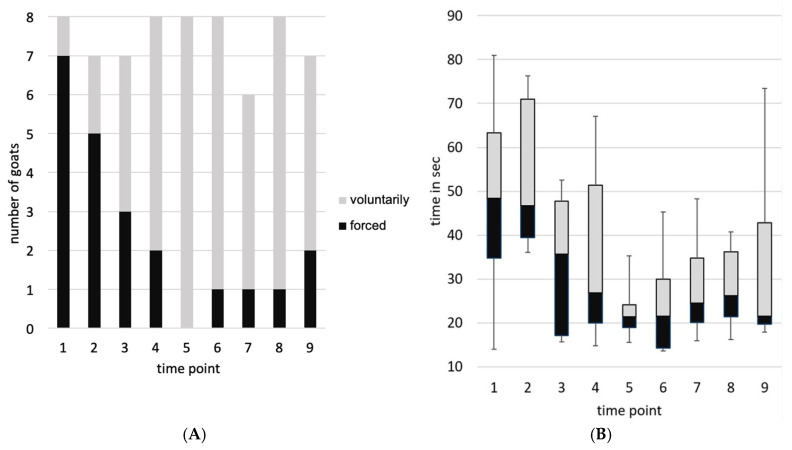
(**A**) Compliance (positioning on the floor target: voluntarily or forced) and (**B**) Latency (time the goats took to enter the position on the floor target) over time.

**Table 1 animals-16-00105-t001:** Summary of training sessions, latency, and achievements. Hand Target: following the hand target; Floor Target: positioning all four hooves on floor target; Touch: tolerating touch while maintaining position on the floor target; Chin on Hand: placing the chin on the trainer’s hand while remaining on the floor target.

Goat	Training Sessions	Duration (min)	Achievement
1	10	120.1	Hand Target (>5 m) Floor Target (without additional signals) + Touch (>5 s) and Chin on Hand (2 s)
2	10	107.7	Hand Target (>5 m) Floor Target (with additional signals) + Touch (>5 s) and Chin on Hand (2 s)
3	8	99.0	Hand Target (>5 m) Floor Target (with additional signals) + Touch (<5 s) and Chin on Hand (7 s)
4	8	92.6	Hand Target (>5 m) Floor Target (with additional signals) + Touch (>5 s) and Chin on Hand (2 s)
5	8	111.7	Hand Target (>5 m) Floor Target (without additional signals) + Touch (<5 s) and Chin on Hand (3 s)
6	7	90.7	Hand Target (>5 m) Floor Target (with additional signals) + Touch (<5 s) and Chin on Hand (1 s)
7	7	95.1	Hand Target (>5 m) Floor Target (with additional signals) + Touch (<5 s) and Chin on Hand (2 s)
8	7	114.2	Hand Target (>5 m) Floor Target (with additional signals) + Touch (<5 s) and Chin on Hand (1 s)

**Table 2 animals-16-00105-t002:** The potential predictor variables and outcome variables.

Designation	Description	Unit/Characteristic
**Predictor Variables**		
Time Point	Time of sampling	1–8/9
Goat	*n* = 8	1–8
Pen	1 (goats 1–4) or 2 (goats 5–8, PFAS exposed)	1 or 2
Temperature	Temperature in pen 1 (goats 1–4) and pen 2 (goats 5–8)	In °C
Humidity	Relative humidity during daily temperature measurement	In %
Person 1	Restraining person	Individual identifier
Person 2	Person taking blood	Individual identifier
**Outcome Variables**		
Cortisol	Cortisol value from blood serum	ng/mL
Thermo	Temperature difference at the medial canthus 5 s after puncturing the skin compared to approx. 3 s before puncturing the skin (T2-T1)	In °C
**Behavior**		
Latency	Duration from entering the sample area to assuming the position on the floor target	In seconds
Compliance	Taking the position for blood collection. Differentiation between “voluntary/lured” (voluntary: floor or hand target; lured: presentation of concentrated feed) and “forced” (led on collar/leash, at least temporarily taut)	“voluntary/lured” OR “forced”
Escape Behavior	Did the goat show intention or action to leave the ground target by shifting the weight towards the exit or moving at least one leg off the ground target?	Yes/No
Defensive Behavior	Did the goat show aversive behavior (either kicking with one leg or hitting the head) during blood collection?	Yes (less than 3 times)/Yes (more than 3 times)/No

## Data Availability

All relevant data are provided upon request by e-mail to carola.fischer-tenhagen@bfr.bund.de.

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
