# Peer review of "Assessing Physiological and Behavioral Stress Parameters in Trained Goats During Repeated Blood Sampling"

_animals, 2025, doi:10.3390/ani16010105_

Round 1
Reviewer 1 Report
Comments and Suggestions for Authors
Great work

Author Response
Dear Reviewer,
thank you for your careful revision of our manuscript. See the attached file for all of our comments and changes.
Kind regards

Reviewer 2 Report
Comments and Suggestions for Authors
Dear Authors,
Please find attached my review report. Hopefully my comments will help you further improve the manusccript.

Author Response

(The authors gave the same response as above.)

Round 2
Reviewer 2 Report
Comments and Suggestions for Authors
Dear Authors,
Please find attached my review report, with some comments that I hope will be helpful in further improving the quality of your work.

Author Response
We sincerely thank the reviewer for their positive feedback and for acknowledging the improvements made to the manuscript. We greatly appreciate the reviewer’s careful evaluation and are pleased that the revisions enhanced the clarity, flow, and overall readability of the study.
Comment 1: L18: “while levels” should be rephrased (however? nonetheless?)
Response 1: Thank you for that comment. We rephrased the line accordingly.
New (Page 1, Simple Summary, Line: 17-19): The results showed that serum cortisol levels and eye temperature did not increase over time. However, the goats approached the blood sampling area faster and more voluntarily as the study progressed.
Comment 2: L266-269: Although there is significant literature on ear, tail, and even body postures as indicators of both positive and negative welfare in ruminants [...] This is just a comment for the authors, intending to support and strengthen their methodological choices. Vocalizations have been studied mainly as negative welfare indicators, and research on goats remains limited. Again, this comment is intended to support the authors’ rationale and choices for not using vocalizations in their study as behavioral indicators. Choosing latency, compliance, escape, and defensive behaviors appear appropriate for assessing animals’ responses to venepuncture. These measures can reflect collaboration and habituation, are relatively feasible to evaluate, and are likely to provide valid indicators of welfare in the relevant context of the study.
Response 2: Thank you for your thoughtful and supportive comment. We appreciate the recognition that our methodological choices are well justified and for the encouraging words that place our work within the broader context of animal welfare research.
Comment 3: L490-2: The Authors mention that “In summary, under well-controlled and positively reinforced handling conditions, repeated exposure to blood sampling as a potential stressful procedure does not necessarily influence stress parameters”. The authors may wish to clarify that repeated exposure does not necessarily increase physiological stress parameters, and/but/while it appears to have a positive effect on the behavioral stress parameters that were used in the study.
Response 3: We agree on your comment and thank you for suggesting to clarify our statement within the conclusion.
New (Page 14, Conclusion, Line: 482-485): In summary, under well-controlled and positively reinforced handling conditions, repeated exposure to blood sampling as a potential stressful procedure does not necessarily influence stress parameters but appears to have a positive effect on the behavioral stress parameters assessed in this study.